

# Nitrous oxide and methane in two tropical estuaries in a peat-dominated region of North-western Borneo

D. Müller[1], H. W. Bange[2], T. Warneke[1], T. Rixen[3,4], M. Müller[5], A. Mujahid[6], and J. Notholt[1,7]

[1] Institute of Environmental Physics, University of Bremen, Otto-Hahn-Allee 1, 28359 Bremen, Germany

5 [2] GEOMAR Helmholtz Centre for Ocean Research Kiel, Düsternbrooker Weg 20, 24105 Kiel, Germany

[3] Leibniz Center for Tropical Marine Ecology, Fahrenheitstr. 6, 28359 Bremen, Germany

[4] Institute of Geology, University of Hamburg, Bundesstr. 55, 20146 Hamburg, Germany

[5] Swinburne University of Technology, Faculty of Engineering, Computing and Science, Jalan Simpang Tiga, 93350 Kuching, Sarawak, Malaysia

10 [6] Department of Aquatic Science, Faculty of Resource Science & Technology, University Malaysia Sarawak, 94300 Kota Samarahan, Sarawak Malaysia

[7] MARUM Center for Marine Environmental Sciences at the University of Bremen, Leobener Str., 28359 Bremen, Germany

*Correspondence to*: D. Müller (dmueller@iup.physik.uni-bremen.de)





**Abstract.** Estuaries are sources of nitrous oxide ($N_2O$) and methane ($CH_4$) to the atmosphere. However, our present knowledge of $N_2O$ and $CH_4$ emissions from estuaries in the tropics is very limited because data is scarce. In this study, we present first measurements of dissolved $N_2O$ and $CH_4$ from two estuaries in a peat-dominated region of north-western Borneo. Two campaigns (during the dry season in June 2013 and during the wet season in March 2014) were conducted in the estuaries of the rivers Lupar and Saribas. Median $N_2O$ concentrations ranged between 7.2 and 12.3 nmol $L^{-1}$ and were higher in the marine end-member ($13.0 \pm 7.0$ nmol $L^{-1}$). $CH_4$ concentrations were low in the coastal ocean ($3.6 \pm 0.2$ nmol $L^{-1}$) and higher in the estuaries (medians between 12.2 and 64.0 nmol $L^{-1}$). The respiration of abundant organic matter and presumably anthropogenic input caused a slight eutrophication, which did not lead to hypoxia or enhanced $N_2O$ concentrations, however. Generally, $N_2O$ concentrations were not related to dissolved inorganic nitrogen concentrations. Thus, the use of an emission factor for the calculation of $N_2O$ emissions from the inorganic nitrogen load leads to an overestimation of the flux from the Lupar and Saribas estuaries. $N_2O$ was negatively correlated with salinity during the dry season, which suggests a riverine source. In contrast, $N_2O$ concentrations during the wet season were not correlated with salinity but locally enhanced within the estuaries, implying that there were additional estuarine sources during the wet (i.e. monsoon) season. Estuarine $CH_4$ distributions were not driven by freshwater input but rather by tidal variations. Both $N_2O$ and $CH_4$ concentrations were more variable during the wet season. We infer that the wet season dominates the variability of the $N_2O$ and $CH_4$ concentrations and subsequent emissions from tropical estuaries. Thus, we speculate that any changes the Southeast Asian monsoon system will lead to changes in the $N_2O$ and $CH_4$ emissions from these systems. We also suggest that the ongoing cultivation of peat soil in Borneo is likely to increase $N_2O$ emissions from these estuaries, while the effect on $CH_4$ remains uncertain.





## 1. Introduction

Nitrous oxide ($N_2O$) and methane ($CH_4$) are greenhouse gases whose global warming potentials exceed that of carbon dioxide ($CO_2$) by far (a factor of 265 for $N_2O$ and 28 for $CH_4$ on a 100 year time horizon, Myhre et al., 2013). Thus, an assessment of the natural and anthropogenic sources and sinks as well as the formation pathways of $N_2O$ and $CH_4$ is essential

to understand present Earth's climate variability and to predict its future development. The world's oceans, including its coastal zones, are sources of $N_2O$ and $CH_4$ to the atmosphere and play a major role in the global budget of atmospheric $N_2O$, but only a minor role in the global budget of atmospheric $CH_4$ (Ciais et al., 2013). Rivers and estuaries are considered hotspots for the production and emission of both $N_2O$ and $CH_4$ (Bange, 2006; Bastviken et al., 2011; Borges et al., 2015; Murray et al., 2015; Seitzinger and Kroeze, 1998).

In aquatic systems, $N_2O$ is mainly formed as a byproduct during nitrification ($NH_4^+ \rightarrow NO_3^-$) and to minor extent as an intermediate during denitrification ($NO_3^- \rightarrow N_2O \rightarrow N_2$) (Freing et al., 2010). In both processes, the yield of $N_2O$ strongly depends on the concentration of ambient oxygen ($O_2$) (e.g. Bange, 2008). Both nitrification and denitrification are microbial processes and occur in the water column and in sediments (e.g. Bange, 2008). $CH_4$ is formed during the decomposition of organic material by microbial methanogenesis (e.g. Reeburgh, 2007; Valentine, 2011; Ferry, 2010). Since $CH_4$ formation

requires strictly anaerobic conditions, it is produced in anoxic environments as found in sediments, in the interior of suspended particles or in zooplankton guts (see e.g. Reeburgh, 2007; Valentine, 2011). Additionally, $CH_4$ is oxidized under aerobic and anaerobic conditions in the water column and in the sediments, respectively (e.g., Valentine, 2011). On the continental shelf so-called geological $CH_4$ can be released directly to the water column through mud volcanoes, via groundwater input or seeping at pockmark structures (e.g., Valentine, 2011). Alternative aerobic $CH_4$ formation pathways in

the nutrient poor (oligotrophic) surface ocean have been suggested (see e.g. Karl et al., 2008; Damm et al., 2010; Zindler et al., 2013), but they are negligible in nutrient rich (eutrophic) coastal systems.

In this study, we present first measurements of dissolved $N_2O$ and $CH_4$ from the estuaries of the rivers Lupar and Saribas, which are located in north-western Borneo (Sarawak, Malaysia, Southeast Asia). These measurements are complemented by dissolved $N_2O$ and $CH_4$ measurements from a cruise on the adjacent coastal shelf of the South China Sea. The Lupar and

Saribas rivers drain catchments which are covered by tropical peatlands, parts of which have been converted to oil palm plantations or other crops. This study aimed at investigating the effect of the carbon-rich peatlands on the $N_2O$ and $CH_4$ estuarine distributions and emissions and the potential impact of estuarine eutrophication. Two intensive sampling campaigns took place during the dry and wet seasons in June 2013 and March 2014, respectively, and provide a first conception of the seasonality in these systems.



## 2. Materials and Methods

### 2.1 Study Area

Our study was conducted in Sarawak, the largest Malaysian state, which is located in the north of the island of Borneo (see Fig. 1a). About 12 % of the area of Sarawak is covered by peatlands (Chai, 2005), approximately 41 % of which have been

converted to oil palm plantations (SarVision, 2011). Sarawak has a tropical climate with a mean annual air temperature of 26.1°C in the capital Kuching and average (1961-1990) annual rainfall of 4101 mm $yr^{-1}$ (DWD, 2007). Between November and February, Sarawak experiences enhanced rainfall due to the Northeast monsoon, while June and July are the driest months (DWD, 2007).

Two macrotidal estuaries which enclose Malaysia's largest peat dome, the Maludam peninsula, are in the focus of this study.

The catchment areas of the Lupar and Saribas rivers are 6558 km² and 1943 km², respectively (Lehner et al., 2006), with approximate discharges of 490 $m^3 s^{-1}$ and 160 $m^3 s^{-1}$ (Müller et al., 2015a). The Lupar catchment comprises mainly the division of Sri Aman, which has a population of approximately 95,000. The Saribas basin belongs largely to the Betong division, with a population of approximately 110,000 (SPU, 2012). Major settlements along the Lupar River and estuary are Sri Aman, Lingga and Sebuyau (Fig. 1b). Important settlements along the Saribas River and estuary are Betong, Pusa and

Beladin. Oil palms are being cultivated in both catchments (Fig. 1b).

In the Lupar and Saribas estuaries, sampling took place during two campaigns in June 2013 and March 2014. Our sampling strategy is described in detail in Müller et al. (2015a). In that study, we showed that precipitation during June 2013 and March 2014 did not deviate much from the historical average, so that we considered the measurements in June 2013 representative of the dry season and those in March 2014 representative of the wet season. Therefore, we refer to the two

campaigns as "MLD dry" (June 2013) and "MLD wet" (March 2014). Note that the peak of the monsoon season is between December and February, so the monsoon recedes in March, which is why our statements about seasonality are conservative.

We complement this dataset with measurements that were performed during the cruise SO218 – SHIVA with the German research vessel (R/V) *Sonne* (15-29 November 2011) (Quack and Krüger, 2013). The cruise started in Singapore and went along the Sarawakian coastline to end in Manila, Philippines (see Fig. 1a). For this study, we selected those samples that

were taken in the coastal ocean off north-western Sarawak (latitude <2.5, 110.5<longitude<111.5) in order to complement our dataset from the Lupar and Saribas estuaries.

### 2.2 Water chemistry

In the Lupar and Saribas inner and outer estuaries, we sampled 20 stations during the dry season and 23 during the wet season. Water samples were taken from approximately 1 m below the water surface. Samples were taken for dissolved

organic carbon (DOC), dissolved inorganic (nitrogenous) nutrients (DIN = $NO_3^-$, $NO_2^-$, and $NH_4^+$), salinity, water temperature and $O_2$:





1. Samples for DOC were filtered (pore size 0.45 μm) and acidified with 21 % phosphoric acid until the pH had dropped below 2. DOC concentrations were determined by high temperature combustion and subsequent measurement of resultant $CO_2$ with a non-dispersive infrared detector (NDIR) (EPA method 415.1). Details are given in Müller et al. (2015a).

2. Inorganic nutrient samples were filtered through a Whatman glass microfibre filter (pore size 0.7 μm), preserved with a mercuric chloride ($HgCl_2$) solution and frozen until analysis. Concentrations of $NO_3^-$, $NO_2^-$ and $NH_4^+$ were determined spectrophotometrically (Hansen and Koroleff, 1999) at a wavelength of 540 nm with a Continuous Flow Analyzer (Alliance, Austria).

3. Salinity and temperature were measured with a CastAway CTD at each station during the MLD cruises.
Additionally, we measured conductivity using a TetraCon 925 conductivity sensor (WTW, Germany). We converted conductivity to salinity using the equations from Bennett (1976). During SO218, salinity and temperature were measured continuously with a thermosalinograph on board.

4. Dissolved oxygen (DO) was measured using a Multi3420 with an FDO 925 oxygen sensor (WTW, Germany).

## 2.3 N₂O and CH₄ measurements

Our setup during the MLD cruises is described in detail in Müller et al. (2015a). Surface water was pumped through a Weiss equilibrator (Johnson, 1999) at a rate of approximately 20 L min$^{-1}$. The headspace air was analyzed using an in-situ Fourier Transform InfraRed (FTIR) trace gas analyzer (University of Wollongong, Australia). This instrument allows for the continuous and simultaneous measurements of several trace gas species, such as $N_2O$ and $CH_4$ as well as $CO_2$ and $CO$ (Müller et al., 2015a) with high accuracy and precision over a wide range of concentrations (Griffith et al., 2012). Spectra
were averaged over 5 min and dry air mole fractions were retrieved using the software MALT5 (Griffith, 1996). The gas dry air mole fractions were corrected for pressure, water and temperature cross-sensitivities (Hammer et al., 2013). The $CO_2$ and CO data obtained from these measurements have been reported by Müller et al., (2015a). Here, we present the $N_2O$ and $CH_4$ measurements. Calibration was achieved by measuring a suite of gravimetrically prepared gas mixtures (Deuste Steininger) ranging from 324 to 3976 ppb $N_2O$ and 1.8 to 239 ppm $CH_4$ in synthetic air, which were calibrated against the World
Meteorological Organization (WMO) reference scale at the Max-Planck-Institute for Biogeochemistry in Jena, Germany. The error associated with the FTIR retrieval is usually small. In ambient air, the total uncertainties reported by Hammer et al. (2013) are 0.084 nmol mol$^{-1}$ for $N_2O$ and 0.25 nmol mol$^{-1}$ for $CH_4$, corresponding to approximately 0.01 % and 0.03 %. The larger source of uncertainty is a potentially remaining disequilibrium between water and headspace in the equilibrator, which can cause an error of <0.2 % for $N_2O$ and 2 % for $CH_4$ (Johnson, 1999).
Water temperature in the equilibrator and in the water as well as ambient air temperature and pressure were monitored as described in Müller et al. (2015a). $N_2O$ and $CH_4$ partial pressures were calculated. Since the sample air was dried before entering the FTIR analyzer, we corrected for the removal of water vapor (Dickson et al., 2007). $N_2O$ molar concentrations were calculated from $N_2O$ fugacity and solubility ($K_0$) according to Weiss and Price (1980). $CH_4$ molar concentrations were



calculated from $CH_4$ partial pressure and solubilities were derived from the equations given by Wiesenburg and Guinasso (1979).

During the R/V *Sonne* cruise SO218, surface sea water was continuously supplied from the ship's hydrographic shaft (moon pool) using a submersible pump at about 4 m water depth. $N_2O$ and $CH_4$ samples were taken in triplicates, preserved with
$HgCl_2$ and analyzed in the lab using headspace equilibration and gas chromatography. Details about the analytic procedures can be found in Walter et al. (2006) and Bange et al. (2010). The average of the three samples was calculated and data was discarded if the standard deviation exceeded 30 % of the average value. For $N_2O$, two additional data points were taken from the surface $N_2O$ concentrations determined in depth profiles.

Atmospheric mixing ratios of $N_2O$ were taken from the Mauna Loa (Hawaii) monitoring station of the NOAA/ESRL
halocarbons in situ program (Dutton et al., 2015) which was the nearest atmospheric $N_2O$ monitoring station in the northern hemisphere. Atmospheric $N_2O$ in Mauna Loa averaged 325.15 ppb in November 2011, 326.26 ppb in June 2013 and 327.08 ppb in March 2014. Atmospheric $CH_4$ was derived from the NOAA/ESRL GMD Carbon Cycle Cooperative Global Air Sampling Network (Dlugokencky et al., 2014). The nearest atmospheric $CH_4$ monitoring station was Bukit Kototabang, Indonesia. Unfortunately, $CH_4$ data from NOAA/ ESRL were not available for March 2014, so we estimated the atmospheric
$CH_4$ during that month from the value reported for March 2013 and an annual growth rate of 4 ppb between 2004-2013 at Bukit Kototabang. Atmospheric $CH_4$ averaged 1841.64 ppb in November 2011, 1798.64 ppb in June 2013 and a value of 1879.35 ppb was derived for March 2014.

## 2.4 Flux estimation

In order to calculate $N_2O$ and $CH_4$ flux densities $F$ (in nmol m$^{-2}$ s$^{-1}$) across the water-air interface, we used the thin film
model, which reads

$$F = kK_0(pGas_{water} - pGas_{air})f, \qquad\qquad (1)$$

where $k$ is the gas exchange velocity (m s$^{-1}$), $K_o$ ist the solubility of $N_2O$ and $CH_4$ in seawater (mol L$^{-1}$ atm$^{-1}$, see Sect. 2.3), $pGas_{water}$ is the partial pressure derived from the equilibrator measurements (natm), $pGas_{air}$ is the partial pressure of the gas in air (natm) as measured at the atmospheric monitoring stations (see Sect. 2.3), and $f$ is a conversion factor from L$^{-1}$ to m$^{-3}$.
For $k$, we used $k_{600}$ values that were reported for the Lupar and Saribas estuaries in Müller et al. (2015a). In-situ $k$ was calculated based on the Schmidt numbers of $N_2O$ and $CH_4$, which relates the kinematic viscosity to the diffusivity of the gas in water. Kinematic viscosity was calculated according to Siedler and Peters (1986), the diffusivity of $N_2O$ was computed using Eq. 2 in Bange et al. (2001) and the diffusivity of $CH_4$ was calculated with the formula given in Jähne et al. (1987). The annual areal flux density was estimated as the average of the wet and dry season values (for the spatial extent that was
covered during the 2013 cruise, see Müller et al., 2015a). The total $N_2O$ and $CH_4$ fluxes (t N yr$^{-1}$ and t C yr$^{-1}$) were calculated assuming an estuarine surface area of 220 km² for the Lupar and 102 km² for the Saribas estuary (Müller et al., 2015a). For $N_2O$, we compare this estimate to one derived using the DIN export and the emission factor for estuaries suggested in Mosier





et al., (1998) (0.0025 kg $N_2O$-N / kg N leaching and runoff ). The DIN export was calculated from river discharge (see Sect. 2.1) and the median DIN concentration.

## 3. Results

### 3.1 Water chemistry

In the estuaries of Lupar and Saribas, salinity ranged from 0-30.6 in the dry season and from 0-31.0 in the wet season. $N_2O$ and $CH_4$ concentration data are available for salinities of 4.3-26.5 (MLD dry) and 6.9-26.4 (MLD wet). For the coastal ocean off north-western Sarawak, $N_2O$ measurements covered salinities between 31.3 and 32.7 and $CH_4$ measurements between 32.2 and 32.7. DO saturation in the estuaries ranged between 63.6 to 94.6 % (MLD dry) and 79.0-100.4 % (MLD wet) (Müller et al., 2015a).

The total DIN concentrations were already published in Müller et al. (2015a). Here, we report the concentrations of the different inorganic nitrogen species. Generally, DIN concentrations were quite low, but locally enhanced. $NO_3^-$ ranged between 6.3 µmol $L^{-1}$ and 36.3 µmol $L^{-1}$ in the dry season and between 2.8 and 17.9 µmol $L^{-1}$ in the wet season. A maximum value of 84.0 µmol $L^{-1}$ was observed during MLD dry approximately 20 km offshore in north-western direction from Sebuyau. There, an influence from the Lupar river plume, but possibly also from the close-by Sadong river was detected

(salinity = 22.2).

$NO_2^-$ concentrations ranged between 0.1 µmol $L^{-1}$ and 0.6 µmol $L^{-1}$ in the dry season and between <0.1 µmol $L^{-1}$ and 2.3 µmol $L^{-1}$ in the wet season. $NH_4^+$ ranged between <0.1 µmol $L^{-1}$ and 2.6 µmol $L^{-1}$ in the dry season, whereas a maximum value of 8.0 µmol $L^{-1}$ was observed at one station in the Lupar estuary. In the wet season, $NH_4^+$ was higher, ranging between 0.2 µmol $L^{-1}$ to 7.8 µmol $L^{-1}$. However, overall, DIN concentrations were higher in the dry season than in the wet season

(Müller et al., 2015a, Table 1).

On the Lupar River, we determined average DIN concentrations of 5.1 µmol $L^{-1}$ (2013) and 5.3 µmol $L^{-1}$ (2014) upstream of the town of Sri Aman, which can be considered unpolluted. This value is 2-6 times lower than the DIN concentrations in the Lupar estuary. On the Saribas River, a slightly enhanced value of 18.6 µmol $L^{-1}$ (2013) was measured at salinity 0 outside the town of Betong, so it cannot be considered unpolluted, yet it was lower than the DIN concentrations in the estuary during

that season (Table 1). We estimated that together, Lupar and Saribas deliver 6086 t N as DIN to the South China Sea every year (Table 1).

### 3.2 $N_2O$

Dissolved $N_2O$ concentrations in the water ranged from 6.9 nmol $L^{-1}$ to 13.4 nmol $L^{-1}$ during MLD dry in June 2013 (corresponding to saturations of 103 to 184 %) and from 6.3 nmol $L^{-1}$ to 116.8 nmol $L^{-1}$ during MLD wet in March 2014 (93

to 1679 % saturation). Most of the time, $N_2O$ was close to atmospheric equilibrium. However, local enhancements were observed: During both the dry and the wet season, the Saribas tributary exhibited markedly higher $N_2O$ concentrations than



the Saribas main river (Fig. 3a,b, Table 2). During the wet season, $N_2O$ concentrations were more variable (higher standard deviation (SD)) and higher maximum concentrations were observed (Table 2). Although the mean $N_2O$ concentrations in the Saribas estuary and Saribas tributary were higher in the wet season as well, no difference was found with regards to the medians (Table 2).

Salinity and $N_2O$ concentrations were correlated in the dry season (r = -0.62), but uncorrelated in the wet season (Fig. 2c). By the town of Sebuyau at the Lupar river mouth, where the Sebuyau river flows into the South China Sea, $N_2O$ was enhanced with concentrations of up to 116.8 nmol L$^{-1}$ in the wet season (not shown in Fig. 2,3). The data from SO218 revealed enhanced $N_2O$ concentrations offshore during November 2011 (see Fig. 2c), with a median of 13.0 ± 7.0 nmol L$^{-1}$ (218 ± 119 % saturation, see Table 2).

$N_2O$ was correlated with DOC (Fig. 2a), whereas this correlation was strong in the dry season (r=0.87) and weak during the wet season (r=0.38). $N_2O$ concentrations were not correlated with DIN (Fig. 2b), $NO_3^-$, $NH_4^+$ (not shown) or DO (Fig. 2d). During the dry season, we found no link between $N_2O$ concentrations and tidal variations (Fig. 4a). In the wet season, however, $N_2O$ exhibited slightly higher concentrations during low tide and its spatiotemporal variation resembled that of $CH_4$ (Fig. 4b).

**3.3 $CH_4$**

Dissolved $CH_4$ concentrations ranged from 5.2 nmol L$^{-1}$ to 59 nmol L$^{-1}$ during MLD dry in 2013 (228 to 2782 % saturation) and from 2.0 nmol L$^{-1}$ to 135 nmol L$^{-1}$ during MLD wet in 2014 (88 to 6003 % saturation) in the Lupar and Saribas estuaries and was spatially variable. The highest value was measured by the town of Sebuyau during MLD wet, whereas the highest median concentrations were detected in the Saribas estuary during the dry season and in the Saribas tributary during the wet

season (Fig. 3c,d; Table 3). In general, no seasonal pattern could be identified. Mean and median $CH_4$ concentrations in the Lupar estuary were comparable in the dry and wet season. In the Saribas estuary, $CH_4$ concentrations were higher during the dry season but on the Saribas tributary, they were higher during the wet season (Table 3). However, higher maximum concentrations were observed in the wet season in both estuaries and the tributary (Table 3).

$CH_4$ concentrations were not correlated with salinity (Fig. 5c). A relatively low marine end-member concentration was

determined during the SO218 cruise: With a median of 3.6 ± 0.2 nmol L$^{-1}$ (176 ± 9 % saturation), $CH_4$ concentrations in the coastal ocean were only slightly enhanced in comparison to the atmospheric equilibrium concentration (see Table 3).

$CH_4$ was not correlated with DOC (Fig. 5a), DO (Fig. 5d), DIN or suspended particulate matter (not shown), but increased with increasing $pCO_2$ (Fig. 5b). This relationship was stronger in the wet season (r=0.55) than in the dry season (r=0.15). One striking feature was that $CH_4$ showed a strong response to the tides (Fig. 4). This is visible for most of the data, even

though we changed location during the measurements. Tidal and spatial variations are overlapping in Fig. 4, but the tidal variation seems to dominate. This is confirmed by one stationary measurement that we conducted over night at one station on the Saribas tributary in 2014 (Fig. 4b): $CH_4$ peaked 4-fold during low tide if compared to the concentration level at high tide.




### 3.4 N$_2$O and CH$_4$ flux densities

Median N$_2$O and CH$_4$ flux densities are listed in Table 4. Both the highest N$_2$O flux density and the highest CH$_4$ flux density were computed for the Saribas tributary. During both seasons, N$_2$O flux densities from the Saribas tributary were up to one order of magnitude higher than from the Lupar and the Saribas estuaries. Annual average N$_2$O flux densities were low for

both Lupar and Saribas, amounting to 1.3 ± 0.3 mmol m$^{-2}$ yr$^{-1}$ and 1.9 ± 1.6 mmol m$^{-2}$ yr$^{-1}$, respectively. The N$_2$O flux density from the Saribas tributary was one order of magnitude higher (12.0 ± 7.5 mmol m$^{-2}$ yr$^{-1}$).

The CH$_4$ flux density observed on the Saribas tributary during the wet season was approximately five times higher than the flux densities from Lupar or Saribas during any season. As a result, the annual average flux from the Saribas tributary (89.2 ± 55.2 mmol m$^{-2}$ yr$^{-1}$) was approximately four times as high as those computed for the Lupar and Saribas estuaries, which

were comparable (22.4 ± 5.4 mmol m$^{-2}$ yr$^{-1}$ and 23.0 ± 19.1 mmol m$^{-2}$ yr$^{-1}$, respectively).

In total, 4 ± 1 t N$_2$O-N yr$^{-1}$ and 59 ± 17 t CH$_4$-C yr$^{-1}$ were emitted from the Lupar estuary and 3 ± 2 t N$_2$O-N yr$^{-1}$ and 28 ± 25 t CH$_4$-C yr$^{-1}$ from the Saribas (Table 4). Using the emission factor of Mosier et al., (1998), we obtained 11 t N$_2$O-N yr$^{-1}$ for the Lupar and 4 t N$_2$O-N yr$^{-1}$ for the Saribas estuary.

## 4. Discussion

### 4.1 Eutrophication in the Lupar and Saribas estuaries

Blackwater rivers and their estuaries usually have very low nutrient concentrations (<1 µmol L$^{-1}$ NO$_3^-$, Kselik and Liong 2004; <5 µmol L$^{-1}$ NO$_3^-$, Alkhatib et al., 2007). Although the Lupar and Saribas rivers are no blackwater rivers, they have several blackwater tributaries (Kselik and Liong 2004; Müller et al., 2015a), and 30.5 % and 35.5 % of their catchments are covered by peat (Müller et al., 2015a). Therefore, rather low nutrient concentrations were expected. However, several

villages and smaller towns are found along the shore of both estuaries, and there is cultivation of sago and oil palm in the catchments (Fig. 1b).

Estuarine DIN concentrations were higher than in the unpolluted freshwater end-member of the Lupar river, indicating that the estuary was indeed slightly eutrophic during the time of our measurements. This eutrophication can be attributed both to the release of DIN during respiration of organic matter, which was shown to be pronounced in the Lupar and Saribas

estuaries (Müller et al., 2015a), especially in the dry season (see discussion below), and to anthropogenic input. These processes were also identified as important sources of inorganic nutrients in the Siak River, a eutrophic blackwater river in central Sumatra, Indonesia (Baum and Rixen, 2014). The DIN concentrations in the Lupar and Saribas estuaries were similar to those reported by Baum and Rixen (2014) for the Siak River.

Eutrophication can lead to enhanced estuarine primary production and consequently to hypoxia. This was not observed.

Although organic matter was respired in the estuaries (Müller et al., 2015a), oxygen depletion was relatively moderate in the surface water due to a quick replenishment from the overlying air, as suggested by the high gas exchange velocity (Table 4).





Since we did not measure oxygen profiles, we can only speculate about the oxygen levels in the bottom water. The high turbulence in the water likely prevented stratification and promoted the ventilation of the water column and, therefore, may have prevented the development of anoxic bottom waters.

## 4.2 $N_2O$

Dissolved $N_2O$ concentrations were mostly close to atmospheric equilibrium concentrations (i.e. 100% saturation), which is in line with other comparable studies in the tropics and subtropics (Richey et al., 1988; Zhang et al., 2010; Rao and Sarma 2013; Borges et al., 2015). High $N_2O$ concentrations were reported for eutrophic and hypoxic coastal waters in the western Indian continental shelf (up to 533 nmol $L^{-1}$, Naqvi et al., 2000) and for the Peruvian upwelling (up to 986 nmol $L^{-1}$, Arevalo-Martinez et al., 2015). Similarly, $N_2O$ concentrations in the subtropical Brisbane estuary were higher than in the

Lupar and Saribas (median concentrations between 7.2 and 12.3 nmol $L^{-1}$), ranging between 9.1 and 45 nmol $L^{-1}$ (Musenze et al., 2014).

The low $N_2O$ concentrations in the Lupar and Saribas estuaries are not surprising. Even though these estuaries were slightly eutrophic, the DIN concentrations were still below the average for tropical non-blackwater rivers (Baum and Rixen, 2014). For estuaries with low DIN concentrations, low $N_2O$ concentrations are expected (Zhang et al., 2010). Interestingly,

enhanced $N_2O$ was measured offshore during cruise SO218. Either, a source of $N_2O$ exists on the continental shelf, or the enhanced $N_2O$ concentrations are due to interannual variability: The SO218 cruise took place in 2011, while the measurements in the Lupar and Saribas estuaries were performed in 2013 and 2014. Ultimately, the SO218 measurements were performed at the onset of the monsoon season, so possibly, seasonal variability plays a role as well. This is in line with the tendency towards higher and more variable $N_2O$ concentrations during MLD wet if compared to MLD dry in the Saribas

tributary.

Stronger oxygen depletion and higher DIN concentrations suggest higher respiration rates during the dry season (Müller et al., 2015a). In accordance with this, $N_2O$ was well correlated with DOC and salinity during the dry season, suggesting that it originated mainly from respiratory activity in freshwater. In contrast, during the wet season, a correlation of $N_2O$ with salinity was not observed and the correlation with DOC was weak, suggesting that additional sources of $N_2O$ existed in the

estuary and obscured these relationships. In the wet season, $N_2O$ showed the same spatial variations as $CH_4$, which were linked to the tidal cycle, which was not the case during MLD dry. This implies that $N_2O$ and $CH_4$ had the same source during the wet season, e.g., production in estuarine sediments or in tidal creeks.

The most striking feature of the spatial distribution of $N_2O$ was its strong variability, with enhanced concentrations in the Saribas tributary during both seasons, indicating a local source. The most obvious candidate for a point source is

anthropogenic, i.e. sewage. However, $NO_3^-$, $NO_2^-$, and $NH_4^+$ were not enhanced in the same way as $N_2O$, suggesting that DIN concentrations are a poor predictor for estuarine $N_2O$ concentrations (see discussion below).



### 4.3 CH₄

Similar to $N_2O$, dissolved $CH_4$ concentrations were relatively moderate. For example, while $CH_4$ concentrations in the partially hypoxic Pearl River estuary ranged between 23 and 2984 nmol $L^{-1}$ (Chen et al., 2008), median $CH_4$ concentrations in our study area varied between 4-64 nmol $L^{-1}$. Similarly, the $CH_4$ concentration range reported by Musenze et al. (2014) for

the Brisbane estuary, Australia, was substantially higher (31-578 nmol $L^{-1}$) than in this study, although both the DOC concentration range and the DO saturation range were similar. This is surprising, as the peat-draining tributaries are extremely oxygen-depleted and contain large amounts of organic matter (Müller et al., 2015b). These conditions are usually suitable for $CH_4$ production. However, the lack of correlation between $CH_4$ and salinity implies that freshwater input was not the main source of $CH_4$ in the estuaries. Since $CH_4$ is a poorly soluble gas, a large fraction might be released to the

atmosphere before reaching the estuary and the coastal ocean. In addition, salt intrusion may inhibit $CH_4$ production in the estuaries. Although $CH_4$ concentrations tended to be higher during the wet season, a clear seasonal pattern of $CH_4$ concentrations was not apparent, which is in line with observations at other tropical (Barnes et al., 2006; Teodoru et al., 2014) and subtropical (Musenze et al., 2014) sites.

The generally positive relationship between $CH_4$ and $pCO_2$ has been observed in other tropical aquatic systems (Teodoru et

al., 2014, Borges et al., 2015) and is indicative of organic matter decomposition as a source of both gases. The strong response of $CH_4$ to tidal variations indicates that $CH_4$ is produced in the sediments and released when the hydrostatic pressure drops during falling tide. The tidal variability of $CH_4$ (and $N_2O$ in the wet season) can also be interpreted as indication of the contribution of intertidal sediments and tidal creeks to the $CH_4$ concentrations in these estuaries. The importance of tidal creeks (Middelburg et al., 2002) and tidal pumping (Barnes et al., 2006; Borges and Abril, 2011) for $CH_4$

concentrations in estuaries is widely recognized.

### 4.4 N₂O and CH₄ flux densities and emissions

With the exception of the Saribas tributary ($F_{annual}$ = 12.0 ± 7.5 mmol $m^{-2}$ $yr^{-1}$), $N_2O$ flux densities were quite low (1.3-1.9 mmol $m^{-2}$ $yr^{-1}$) and at the lower end of the range reported for Indian estuaries (-0.4-5.2 mmol $m^{-2}$ $yr^{-1}$, Rao and Sarma, 2013). $CH_4$ flux densities (22.4-89.2 mmol $m^{-2}$ $yr^{-1}$) were within the ranges reported for other tropical sites: Koné et al. (2010), for

example, determined flux densities of 28.5-123.4 mmol $m^{-2}$ $yr^{-1}$ for stratified lagoons of Ivory Coast; Shalini et al. (2006) report flux densities of 19.7-102.2 mmol $m^{-2}$ $yr^{-1}$ for Pulicat lake, India; and Biswas et al. (2007) measured $CH_4$ flux densities between 0.7 and 49 mmol $m^{-2}$ $yr^{-1}$ in the estuaries of the Sundarban mangrove ecosystem. However, we caution that the comparison suffers from the different approaches to determining the gas exchange velocity $k$. Koné et al. (2010), Shalini et al. (2006), Biswas et al. (2007) and Rao and Sarma (2013) used empirical equations relating $k$ to wind speed, while we

used estimates based on floating chamber measurements. We refer to our discussion in Müller et al. (2015a), where we showed that our estimates derived by floating chamber measurements yielded higher values than if we had used empirical relationships with wind speed. We argued that floating chamber measurements offered a better representation of the actual

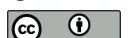



flux than gas exchange models that rely on empirical relationships with wind speed, which were initially derived for the open ocean (Wanninkhof, 1992) and do not consider current-induced turbulence as a driver of gas exchange. Musenze et al. (2014) used empirical models for both the wind-related and the current-related gas exchange velocity and added them up to derive flux estimates for the Brisbane estuary. Not surprisingly, the fluxes they report were much higher than in the

aforementioned studies, with 0.8-28.0 mmol $N_2O$ $m^{-2}$ $yr^{-1}$ and 7.5 to 636 mmol $CH_4$ $m^{-2}$ $yr^{-1}$. Compared with these estimates, the $N_2O$ and $CH_4$ fluxes from Lupar and Saribas are quite low, which is consistent with the lower $N_2O$ and $CH_4$ concentrations as discussed above.

On the basis of our floating chamber measurements (see Müller et al, 2015a) we computed an overall annual flux of 7 t $N_2O$-N for the Lupar and Saribas estuaries. This is only half of the estimate (15 t $N_2O$-N $yr^{-1}$) computed using the Mosier et al.

(1998) emission factor. The reason is obviously the missing relationship between DIN concentrations and $N_2O$ emissions in the studied system. Although across estuaries, increasing DIN is generally associated with increasing $N_2O$ (Murray et al., 2015), the relationship is not linear: Zhang et al. (2010) found a polynomial relationship between DIN and $N_2O$ across estuaries in different geographic locations. The emission factor from Mosier et al. (1998), in contrast, is based on constant $N_2O$/DIN ratio. Therefore, it is not surprising that for low DIN environments, as our study site, the use of emission factors

leads to a considerable bias. In line with the data presented here, Borges et al. (2015) did not find a relationship between $N_2O$ and DIN from measurements in several tropical and sub-tropical African rivers and estuaries. This could indicate that this relationship is not applicable in tropical systems and demonstrates the need of further studies in low latitudes to improve estimates of global $N_2O$ emissions from estuaries.

### 4.5 Implications for future land use change

Nutrient loads to estuaries are predicted to increase in the future in most of Southeast Asia (Seitzinger and Kroeze, 1998). Our results imply that eutrophication in peat-dominated estuaries does not automatically lead to enhanced $N_2O$ emissions. However, the localized elevation of $N_2O$ concentrations, as observed on the Saribas tributary, suggests that estuarine $N_2O$ concentrations might be impacted by local anthropogenic sources. Indeed, it has been shown that $N_2O$ emissions from peat soils depend on land use and that cultivated sites generally exhibit higher $N_2O$ fluxes to the atmosphere (Hadi et al., 2000). It

is likely that this behavior is mirrored in the aquatic systems as well.

As $CH_4$ emissions from peatlands depend largely on the water table (Couwenberg et al., 2010), $CH_4$ fluxes from peatlands are enhanced under sago (Melling et al., 2005) and rice (Couwenberg et al., 2010) and reduced under oil palm (Melling et al., 2005). The cultivation of sago in the catchment of the Saribas tributary might therefore partially explain the observed high $CH_4$ fluxes from the Saribas tributary. The future development of $CH_4$ dynamics in estuaries in this region in

dependence of land use change is hard to predict, as there are potentially counteracting effects of the conversion of peatlands to oil palm and other industrial crops, which require different agricultural practices (flooding versus drainage).





## 5. Conclusions

Overall, we found that the two tropical estuaries of the rivers Lupar and Saribas in a peat-dominated region in Malaysia were small to moderate sources of $N_2O$ and $CH_4$ to the atmosphere. DIN concentrations were slightly enhanced compared to the unpolluted riverine end-member. This eutrophication did not lead to hypoxia or to enhanced $N_2O$ concentrations. DIN was generally a poor predictor of $N_2O$, which provides further evidence that the use of emission factors for the calculation of $N_2O$ fluxes from tropical estuaries is inappropriate. Although predictions about the future development of the Southeast Asian monsoon are highly uncertain and locally variable, chances are that rainfall will moderately increase in this region (Christensen et al., 2013). Our results suggest that this may increase the $N_2O$ and probably also the $CH_4$ emissions from estuaries, as the concentrations of both gases were more variable during the wet season, with higher maximum concentrations of both gases and additional sources of $N_2O$ in the estuaries. For our study area, additional sampling at the peak of the monsoon season would be desirable in order to consolidate these statements. Yet, our results provide a first conception of the seasonality in these systems and underline the fact that time series measurements (seasonal sampling) are vital for the understanding of $N_2O$ and $CH_4$ fluxes from aquatic systems in monsoonal regions.

### Acknowledgments

We would like to thank the Sarawak Biodiversity Center for permission to conduct research in Sarawak waters (Permit No. SBC-RA-0097-MM and export permit SBC-EP-0040-MM). We thank Hella van Asperen (University of Bremen, Germany), Nastassia Denis, Felicity Kuek, Joanne Yeo, Hong Chang Lim, Edwin Sia (all Swinburne University, Malaysia) and all scientists and students from Swinburne University and the University of Malaysia Sarawak who were involved in the MLD cruises and their preparation. Lukas Chin and the crew members of the SeaWonder are acknowledged for their support. We thank Franziska Wittke (GEOMAR), who performed the sampling for $N_2O$ and $CH_4$ on the R/V *Sonne* cruise SO218 and Annette Kock (GEOMAR), who computed the data from SO218. The authors thank Matthias Birkicht and Dorothee Dasbach (ZMT Bremen, Germany) for their help in the lab performing the analyses of the MLD samples. We acknowledge the University of Bremen for funding the MLD cruises through the "exploratory project" in the framework of the University's Institutional Strategy and the EU FP7 project InGOS for supporting the development of the FTIR measurements. Cruise SO218 was supported by the EU FP7 project SHIVA under grant agreement no. 226224.



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




**Tables**

| | | DIN (µmol L$^{-1}$) | | | | | N export (t yr$^{-1}$) |
|---|---|---|---|---|---|---|---|
| | | **Min** | **Max** | **Mean** | **Median** | **SD** | |
| **Lupar** | **dry** | 20.9 | 30.8 | 23.6 | 22.2 | 3.6 | 4532 |
| | **wet** | 11.9 | 21.7 | 18.2 | 19.7 | 3.2 | |
| **Saribas** | **dry** | 25.5 | 37.6 | 31.0 | 30.0 | 6.1 | 1554 |
| | **wet** | 10.0 | 18.0 | 14.0 | 14.0 | 5.6 | |
| **Saribas tributary** | **dry** | 20.5 | 30.0 | 24.3 | 22.4 | 5.0 | n.d. |
| | **wet** | 10.9 | 18.2 | 13.0 | 11.4 | 3.5 | |

Table 1: Estuarine DIN concentration medians for the spatial extent that was covered in 2013 and the estimated inorganic N export to the South China Sea.





| | | Concentrations (nmol L$^{-1}$) | | | | | Saturations (%) | | | | |
|---|---|---|---|---|---|---|---|---|---|---|---|
| | | Min | Max | Mean | Median | SD | Min | Max | Mean | Median | SD |
| **Lupar** | **dry** | 7.2 | 9.3 | 7.7 | 7.6 | 0.4 | 106 | 116 | 108 | 108 | 1 |
| | **wet** | 6.3 | 13.8 | 7.7 | 7.2 | 1.0 | 93 | 208 | 116 | 109 | 15 |
| **Saribas** | **dry** | 7.3 | 8.4 | 7.9 | 7.9 | 0.2 | 110 | 118 | 114 | 113 | 2 |
| | **wet** | 8.2 | 19.3 | 9.0 | 8.7 | 1.2 | 118 | 592 | 134 | 126 | 46 |
| **Saribas** | **dry** | 8.1 | 13.4 | 11.8 | 12.3 | 1.5 | 117 | 184 | 165 | 169 | 19 |
| **tributary** | **wet** | 9.1 | 24.4 | 13.5 | 11.9 | 4.3 | 130 | 340 | 189 | 169 | 59 |
| **Coastal ocean** | | 10.1 | 27.3 | 15.8 | 13.0 | 7.0 | 168 | 462 | 266 | 218 | 119 |

Table 2: N$_2$O concentrations and saturations in the Lupar, Saribas and the Saribas tributary and in the coastal ocean. Values for the estuaries are given for the spatial extent of the rivers that was covered in 2013.





| | | Concentrations (nmol L$^{-1}$) | | | | | Saturations (%) | | | | |
|---|---|---|---|---|---|---|---|---|---|---|---|
| | | Min | Max | Mean | Median | SD | Min | Max | Mean | Median | SD |
| **Lupar** | **dry** | 7.3 | 42.3 | 18.7 | 10.6 | 12.2 | 339 | 1910 | 837 | 486 | 520 |
| | **wet** | 2.0 | 61.2 | 24.3 | 13.5 | 20.3 | 88 | 2799 | 1100 | 614 | 915 |
| **Saribas** | **dry** | 8.3 | 58.8 | 28.2 | 25.6 | 15.5 | 397 | 2782 | 1331 | 1188 | 727 |
| | **wet** | 9.0 | 68.5 | 15.1 | 12.2 | 9.1 | 401 | 3050 | 671 | 545 | 395 |
| **Saribas tributary** | **dry** | 5.2 | 53.3 | 20.6 | 12.4 | 16.9 | 228 | 2458 | 942 | 554 | 784 |
| | **wet** | 23.3 | 113.9 | 63.2 | 64.0 | 27.4 | 1040 | 5058 | 2773 | 2786 | 1198 |
| **Coastal ocean** | | 3.3 | 3.7 | 3.5 | 3.6 | 0.2 | 166 | 188 | 177 | 176 | 9 |

Table 3: CH$_4$ concentrations and saturations in the Lupar, Saribas and the Saribas tributary and in the coastal ocean. Values for the estuaries are given for the spatial extent of the rivers that was covered in 2013.





| | | $k_{600}$ (cm h$^{-1}$) | Flux density (nmol m$^{-2}$ s$^{-1}$) | | Annual average flux density (mmol m$^{-2}$ yr$^{-1}$) | | Flux (t N yr$^{-1}$ and t C yr$^{-1}$) | |
|---|---|---|---|---|---|---|---|---|
| | | | N$_2$O | CH$_4$ | N$_2$O | CH$_4$ | N$_2$O | CH$_4$ |
| **Lupar** | **dry** | n.d. | 0.04 ± 0.01 | 0.59 ± 0.14 | 1.3 ± 0.3 | 22.4 ± 5.4 | 4 ± 1 | 59 ± 17 |
| | **wet** | 20.5 ± 4.9 | 0.04 ± 0.01 | 0.83 ± 0.20 | | | | |
| **Saribas** | **dry** | n.d. | 0.04 ± 0.03 | 1.01 ± 0.84 | 1.9 ± 1.6 | 23.0 ± 19.1 | 3 ± 2 | 28 ± 25 |
| | **wet** | 13.2 ± 11.0 | 0.08 ± 0.07 | 0.45 ± 0.37 | | | | |
| **Saribas tributary** | **dry** | n.d. | 0.39 ± 0.24 | 0.81 ± 0.50 | 12.0 ± 7.5 | 89.2 ± 55.2 | n.d. | n.d. |
| | **wet** | 23.9 ± 14.8 | 0.37 ± 0.23 | 4.84 ± 3.00 | | | | |

Table 4: $k_{600}$ values and median N$_2$O and CH$_4$ areal and total fluxes from the Lupar, Saribas and the Saribas tributary. The uncertainties relates to the maximum variability of the $k_{600}$ value. Values are given for the spatial extent of the rivers that was covered in 2013.



**Figures**



Figure 1: Map of the study area showing (a) the location of Sarawak on the island of Borneo and the cruise track of the R/V *Sonne* in November 2011 (SO218). (b) Close-up map of the Lupar and Saribas estuaries, enclosing the Maludam peninsula, showing the major settlements along the rivers and estuaries. Peat areas are indicated by the dark grey area, oil palm plantations as of CIFOR (2014) are shown in red.



Figure 2: Relationship of nitrous oxide concentrations with (a) dissolved organic carbon (DOC), (b) dissolved inorganic nitrogen (DIN), (c) salinity and (d) dissolved oxygen (DO). MLD refers to the campaigns on the Lupar and Saribas estuaries in the dry and wet season, respectively, SO218 denotes data from the R/V *Sonne* cruise.



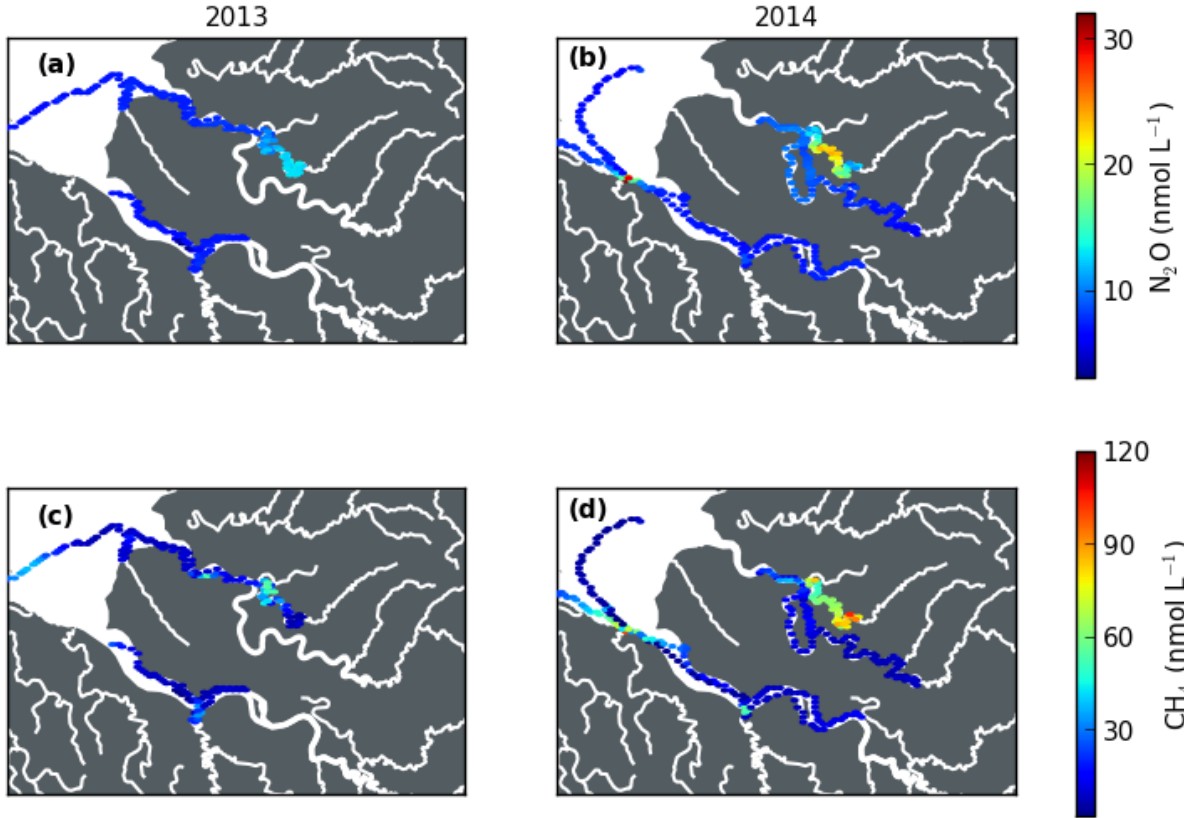

Figure 3: Dissolved $N_2O$ (a+b) and $CH_4$ (c+d) concentrations measured during the 2013 (left) and 2014 (right) MLD campaigns.



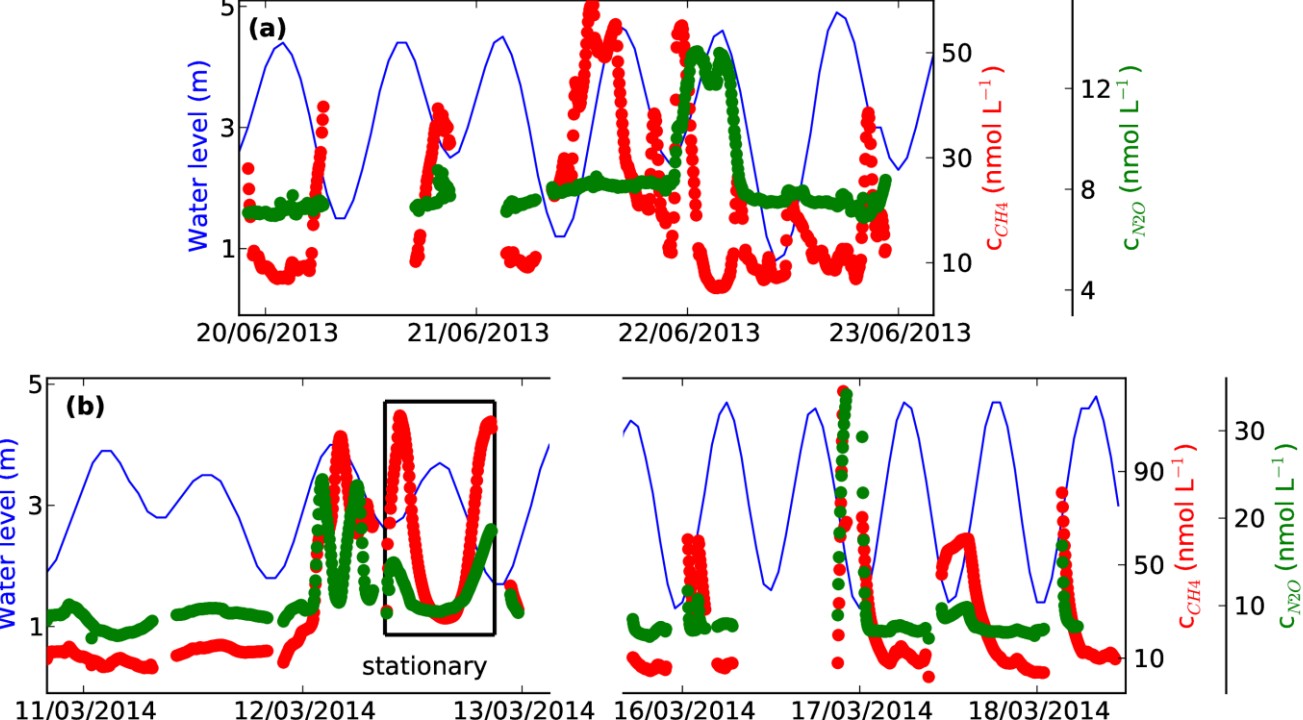

Figure 4: Time series of methane and nitrous oxide concentrations in the Lupar and Saribas estuaries measured during the dry season (2013) campaign (upper panel) and the wet season campaign (lower panel) and the water level as predicted for Pulau Lakei (+1 h for Lupar and Saribas). Spatial, temporal and tidal variations are overlapping in the Figure. One stationary measurement, as recorded on the Saribas river in 2014, is denoted with the black box. Note the discontinuous time axis in the lower panel.






Figure 5: Relationship of methane concentrations with (a) dissolved organic carbon (DOC), (b) $pCO_2$, (c) salinity and (d) dissolved oxygen (DO). MLD refers to the campaigns on the Lupar and Saribas estuaries in the dry and wet season, respectively, SO218 denotes data from the R/V *Sonne* cruise.