# Peer review of "Nitrous oxide and methane in two tropical estuaries in a peatdominated region of North-western Borneo"

_Biogeosciences, 2016_

## Referee Comment (RC1) · Anonymous Referee #1 · 19 Feb 2016

This paper describes CH4 and N2O distributions in two tropical estuaries. Current literature for estuarine CH4 and N2O concentrations is still limited, and this type of study is significant for our scientific understanding and relevant to BG. The paper is well written and reads easily. However, there are several issues that need to be addressed prior to publication.

1) The dataset of CH4 and N2O concentrations does not cover the region of low salinity (0-5), where high CH4 and N2O might be expected. Furthermore, no sample was taken at the river-end member. Hence it is hard to take a full picture of trace gas variation in the whole estuary system.

2) Page 5, line 13: Does the DO sensor calibrated with Winkler titration method?

[Figure]

3) Page 6, line 25: I think the authors should give a little more detail about the k600 they used. For example, were they measured in situ or calculated by widely used wind-speed related relationships in the literature?

4) Table 3 showed that dissolved CH4 in the Lupar estuary was low and undersaturated (2nM and 88%). What's possible reasons for this?

5) Table 4 didn't show k600 for dry season, and the authors should explain what k600 was used for the calculation for dry season. k600 for Saribas tributary and Lupar estuary are almost twice of that for Saribas. Do the authors have any idea of the reason?

6) Figure 1: Scales should be added and the South China Sea should be located on the map.

7) Figure 2: N2O vs salinity, it was shown that there are great N2O peaks during wet seasons between the areas with salinity of 12-15, suggesting a significant N2O source. The authors should discuss this in the text.

---

## Referee Comment (RC2) · Anonymous Referee #2 · 24 Feb 2016

This paper presents new information on seasonal differences in N2O and CH4 fluxes in mangrove environments. It furthers our understanding of the role of natural factors such as salinity, DO and DOC on greenhouse gas production in a generally sparsely-sampled region. There are a few minor issues to address:

Spelling/Grammar: This paper is very well written, there were only a few issues that I could find.

1. Page 8, line 10: "N2O was correlated with DOC (Fig. 2a), whereas this correlation was strong..." I think this should be "N2O was correlated with DOC (Fig. 2a), *and* this correlation was strong..." or just "N2O was correlated with DOC (Fig. 2a); this correlation was strong..."

2. Page 9, like 17: "...the Lupar and Saribas rivers are no blackwater rivers..." Is this meant to say "...are *not* blackwater rivers..." It's technically correct either way, but the first phrasing sounds more colloquial.

3. Page 10, line 15: "Either, a source of N2O exists on the continental shelf..." This comma is unnecessary.

Other Comments:

1. This paper refers to k-value calculations derived from floating chamber experiments (covered in an earlier paper, Müller et al., 2015). It might be a good idea to make it clear in the methods section that these k values are from floating chambers, to distinguish this k calculation from the more common technique of estimating k-values from equations well-known in the literature.

2. On a related point, gas transfer velocities can be temporally and spatially heterogeneous. Were the floating chamber measurements made near the field site? Were they made upstream, downstream, or along the length of the estuaries? A brief mention of the location or timing of the floating chamber measurements might give a better idea of the precision of this approach to k-value calculation.

3. The authors thoroughly document the source of the atmospheric mixing ratios of N2O and CH4 (which are needed in order to calculate water-to-air fluxes). However, from what I understand, the N2O flux totals are likely going to be much more sensitive to the choice of k-value. Is there some reason why the k-value would be particularly sensitive to the atmospheric mixing value (say if it were 325.25 ppb rather than 325.15 ppb)? Considering that the local air mixing ratio could be slightly greater or less than the Mauna Loa value, it might be good to mention that this isn't a large source of error in the calculation.

---

## Author Comment (AC1) · 23 Mar 2016

We thank Anonymous Referee 2 for the positive evaluation of our manuscript. Our replies to the detailed comments can be found below.

**This paper presents new information on seasonal differences in N2O and CH4 fluxes in mangrove environments. It furthers our understanding of the role of natural factors such as salinity, DO and DOC on greenhouse gas production in a generally sparsely-sampled region. There are a few minor issues to address:**

**Spelling/Grammar: This paper is very well written, there were only a few issues that I could find.**

**1. Page 8, line 10: "N2O was correlated with DOC (Fig. 2a), whereas this correlation was strong..." I think this should be "N2O was correlated with DOC (Fig. 2a), \*and\* this correlation was strong..." or just "N2O was correlated with DOC (Fig. 2a); this correlation was strong..."**
This will be corrected in the revised manuscript (suggestion 2).

**2. Page 9, like 17: "...the Lupar and Saribas rivers are no blackwater rivers..." Is this meant to say "...are \*not\* blackwater rivers..." It's technically correct either way, but the first phrasing sounds more colloquial.**
We will change this to the more formal phrasing in the revised manuscript.

**3. Page 10, line 15: "Either, a source of N2O exists on the continental shelf..." This comma is unnecessary.**
The comma will removed from the sentence in the revised manuscript.

**Other Comments:**
**1. This paper refers to k-value calculations derived from floating chamber experiments (covered in an earlier paper, Müller et al., 2015). It might be a good idea to make it clear in the methods section that these k values are from floating chambers, to distinguish this k calculation from the more common technique of estimating k-values from equations well-known in the literature.**
We agree, we will replace the following sentence:
*For k, we used k600 values that were reported for the Lupar and Saribas estuaries in Müller et al. (2015a).*

with

*For k, we used k600 values that were derived for the Lupar and Saribas estuaries using the floating chamber method (Müller et al., 2015a). Floating chamber measurements were conducted at several locations along the estuaries during the wet season campaign and averaged over the spatial extent of the individual estuaries. We argued in Müller et al. (2015a) that the k600 values determined in this way are more appropri-*

*ate than commonly used wind speed parameterizations, which neglect the influence of tidal currents and the water flow velocity.*

**2. On a related point, gas transfer velocities can be temporally and spatially heterogeneous. Were the floating chamber measurements made near the field site? Were they made upstream, downstream, or along the length of the estuaries? A brief mention of the location or timing of the floating chamber measurements might give a better idea of the precision of this approach to k-value calculation.** We will add this information in the methods section as indicated in the reply to your comment above.

**3. The authors thoroughly document the source of the atmospheric mixing ratios of N2O and CH4 (which are needed in order to calculate water-to-air fluxes). However, from what I understand, the N2O flux totals are likely going to be much more sensitive to the choice of k-value. Is there some reason why the k-value would be particularly sensitive to the atmospheric mixing value (say if it were 325.25 ppb rather than 325.15 ppb)? Considering that the local air mixing ratio could be slightly greater or less than the Mauna Loa value, it might be good to mention that this isn't a large source of error in the calculation.** The k600 values were derived from $CO_2$ fluxes measured with a floating chamber and simultaneous measurements of the $pCO_2$ in water and in the air. That is, when we determined the k600 in the first place, we used atmospheric concentrations measured at the site and not mixing ratios determined elsewhere. Therefore, for the k600 values themselves, the error for $pCO2^{air}$ is not significantly larger than the instrument's uncertainty. Nevertheless, as you mentioned in comment 2 as well, the k600 can be temporally and spatially variable. Therefore, with respect to the calculated $N_2O$ and $CH_4$ fluxes, k600 still causes the largest uncertainty. We did a simple error propagation calculation on the average values, assuming an error of 60% for k, 0.5% for $K_0$, 2% for $pCH_4^{water}$ and 0.2% for $pN_2O^{water}$ (disequilibrium error, Johnson 1999) and 1% for $pCH_4$ and $pN_2O$ in air. As a result, the uncertainty in k accounts for 96% of the flux uncertainty. Therefore, we think our approach of reporting the flux with the % uncertainty of k600 is justified. We will change the caption of Table 4 to

*k600 values and median $N_2O$ and $CH_4$ areal and total fluxes from the Lupar, Saribas and the Saribas tributary. The uncertainties relate to the maximum variability of the k600 value, as the k600 uncertainty propagation was responsible for approximately 96% of the flux uncertainty. Values are given for the spatial extent of the rivers that was covered in 2013.*

---

## Author Response (AR1)

We thank Anonymous Referee 1 for his helpful comments and suggestions. Our detailed answers can be found below.

**This paper describes CH4 and N2O distributions in two tropical estuaries. Current literature for estuarine CH4 and N2O concentrations is still limited, and this type of study is significant for our scientific understanding and relevant to BG. The paper is well written and reads easily. However, there are several issues that need to be addressed prior to publication.**

**1) The dataset of CH4 and N2O concentrations does not cover the region of low**

[Figure]

**salinity (0-5), where high CH4 and N2O might be expected. Furthermore, no sample was taken at the river-end member. Hence it is hard to take a full picture of trace gas variation in the whole estuary system.**

The correlation of $N_2O$ with salinity during the dry season indicates that the freshwater end-member might indeed exhibit the highest $N_2O$ during the dry season. We calculated the expected river end-member using the correlation of $N_2O$ with salinity during the dry season and found concentrations of 9.1 nM, (Lupar, r = 0.5), 9.3 nM (Saribas, r=0.8) and 15 nM (Saribas tributary, r=0.9). For $CH_4$, this is a bit harder to do, as we did not observe a correlation between methane and salinity. While we agree that the lack of data in the low salinity region deserves a more thorough discussion, which we will include in the revised manuscript in sections 4.2 and 4.3, our main conclusions can be maintained despite the lack of data for the upper estuaries:

1) Eutrophication did not lead to enhanced $N_2O$. - Data from the river end-member would not provide additional information, as eutrophication was not observed in the river end-member.

2) DIN was a poor predictor of $N_2O$. - This was mainly inferred from Figure 2b), a river end-member data point is not likely to change this overall observation.

3) Postulation of additional $N_2O$ and $CH_4$ sources during the wet season. - This conclusion is based mainly on the observation of high $N_2O$ and $CH_4$ values at salinities 10-20. Of course, it would be extremely interesting to see if a similar observation could be made in the freshwater region.

**2) Page 5, line 13: Does the DO sensor calibrated with Winkler titration method?**
The FDO 925 sensor was calibrated by the manufacturer (WTW, Germany). According to the manufacturer, user calibration is not required for the specified lifetime of the sensor. Nevertheless, a routine function check was performed in water vapor saturated air, using the check and calibration vessel (FDO (R) check) that was provided with the sensor. This information will be added in the revised manuscript.

[Figure]

**3) Page 6, line 25: I think the authors should give a little more detail about the k600 they used. For example, were they measured in situ or calculated by widely used wind-speed related relationships in the literature?**

The k600 that we used were derived from $CO_2$ fluxes measured with a floating chamber in 2014. The measurements are described in detail in Müller et al., 2015a. From simultaneous measurements of the $CO_2$ flux and the water and air $pCO_2$, we derived $kCO_2$ and, ultimately, k600 for these estuaries. In the revised manuscript, we will change the following sentence:

*For k, we used k600 values that were reported for the Lupar and Saribas estuaries in Müller et al. (2015a).*

to

*For k, we used k600 values that were derived for the Lupar and Saribas estuaries using the floating chamber method (Müller et al., 2015a). Floating chamber measurements were conducted at several locations along the estuaries during the wet season campaign and averaged over the spatial extent of the individual estuaries. We argued in Müller et al. (2015a) that the k600 values determined in this way are more appropriate than commonly used wind speed parameterizations, which neglect the influence of tidal currents and the water flow velocity.*

**4) Table 3 showed that dissolved CH4 in the Lupar estuary was low and undersaturated (2nM and 88%). What's possible reasons for this?**

This seems to be an artifact. The reasons why we think so are the following: 1.) There are only two datapoints directly following each other which indicate undersaturation. 2.) With the exception of CO, all gas concentrations that we could retrieve from the respective spectra indicate undersaturation or atmospheric equilibrium values ($N_2O$, $CH_4$ and $CO_2$). This indicates that atmospheric air was in the measurement cell. 3.) No measurements were taken during the 20 minutes preceding the undersaturation values. A background measurement (empty cell) had been performed before that, and

the cell was filled with atmospheric air afterwards before the measurements of equilibrator air continued. If we reconsider these facts, it seems likely that atmospheric air was still in the cell. We would therefore exclude the two data points from the analysis. The numbers will be corrected in the revised manuscript.

**5) Table 4 didn't show k600 for dry season, and the authors should explain what k600 was used for the calculation for dry season. k600 for Saribas tributary and Lupar estuary are almost twice of that for Saribas. Do the authors have any idea of the reason?**
We used the same k600 for the dry and wet season. In the revised manuscript, we will add a more detailed description of the k600 values used in the methods section (see comment 3).
We discussed the variability of the k600 values in Müller et al., 2015a and after adding more detailed information in the methods section, the reader will be referred to that publication. There, we reasoned that the strong currents in the Lupar estuary are responsible for the relatively higher k600. The higher gas exchange velocity in the Saribas tributary if compared to the Saribas is consistent with the notion that the gas exchange velocity decreases with increasing stream order (Raymond et al., 2012).

**6) Figure 1: Scales should be added and the South China Sea should be located on the map.**
The Figure will be revised accordingly.

**7) Figure 2: N2O vs salinity, it was shown that there are great N2O peaks during wet seasons between the areas with salinity of 12-15, suggesting a significant N2O source. The authors should discuss this in the text.**
On page 10, lines 23-27 of the discussion paper, we tried to combine several lines of evidence as to where this $N_2O$ comes from. Our argument is as follows. 1) There are sources in the estuary (i.e., salinities 10-20) in the wet season which we did not observe in the dry season. 2) Another feature of $N_2O$ that we observed in the wet but not in the dry season was the co-variation of $N_2O$ with $CH_4$. 3) In the wet season,

both $N_2O$ and $CH_4$ varied with the tides, with higher concentrations during low tide. We inferred from 1) that there are additional sources of $N_2O$ in the wet season, from 2) that they are sources of both $N_2O$ and $CH_4$ and from 3) that either tidal creeks or the estuarine sediments constitute this additional source. We will modify the sections 4.2 and 4.3 in the revised manuscript in order to provide a more specific and detailed discussion.

[revised manuscript text omitted]

Comment [d1]: Figure 1 was revised (scale bars were inserted and the South China Sea was labelled)

Figure 1: Map of the study area showing (a) the location of Sarawak on the island of Borneo and the cruise track of the R/V *Sonne* in November 2011 (SO218). (b) Close-up map of the Lupar and Saribas estuaries, enclosing the Maludam peninsula, showing the major settlements along the rivers and estuaries. Peat areas are indicated by the dark grey area, oil palm plantations as of CIFOR (2014) are shown in red.

[Figure]

[Figure]

Figure 2: Relationship of nitrous oxide concentrations with (a) dissolved organic carbon (DOC), (b) dissolved inorganic nitrogen (DIN), (c) salinity and (d) dissolved oxygen (DO). MLD refers to the campaigns on the Lupar and Saribas estuaries in the dry and wet season, respectively, SO218 denotes data from the R/V *Sonne* cruise.

[Figure]

Figure 3: Dissolved N$_2$O (a+b) and CH$_4$ (c+d) concentrations measured during the 2013 (left) and 2014 (right) MLD campaigns.

[Figure]

[Figure]

Figure 4: Time series of methane and nitrous oxide concentrations in the Lupar and Saribas estuaries measured during the dry season (2013) campaign (upper panel) and the wet season campaign (lower panel) and the water level as predicted for Pulau Lakei (+1 h for Lupar and Saribas). Spatial, temporal and tidal variations are overlapping in the Figure. One stationary measurement, as recorded on the Saribas River in 2014, is denoted with the black box. Note the discontinuous time axis in the lower panel.

[Figure]

[Figure]

Figure 5: Relationship of methane concentrations with (a) dissolved organic carbon (DOC), (b) $pCO_2$, (c) salinity and (d) dissolved oxygen (DO). MLD refers to the campaigns on the Lupar and Saribas estuaries in the dry and wet season, respectively, SO218 denotes data from the R/V *Sonne* cruise.